# Thermal Flow Meter with Integrated Thermal Conductivity Sensor

**DOI:** 10.3390/mi14071280

**Published:** 2023-06-21

**Authors:** Shirin Azadi Kenari, Remco J. Wiegerink, Henk-Willem Veltkamp, Remco G. P. Sanders, Joost C. Lötters

**Affiliations:** 1Integrated Devices and Systems Group (IDS), University of Twente, 7522 NB Enschede, The Netherlands; r.j.wiegerink@utwente.nl (R.J.W.); r.g.p.sanders@utwente.nl (R.G.P.S.); j.c.lotters@utwente.nl (J.C.L.); 2MESA+ Institute, 7522 NH Enschede, The Netherlands; h.veltkamp@utwente.nl; 3Bronkhorst High-Tech BV, 7261 AK Ruurlo, The Netherlands

**Keywords:** thermal flow sensor, thermal conductivity, calorimetric sensor, Wheatstone bridge

## Abstract

This paper presents a novel gas-independent thermal flow sensor chip featuring three calorimetric flow sensors for measuring flow profile and direction within a tube, along with a single-wire flow independent thermal conductivity sensor capable of identifying the gas type through a simple DC voltage measurement. All wires have the same dimensions of 2000 μm in length, 5 μm in width, and 1.2 μm in thickness. The design theory and COMSOL simulation are discussed and compared with the measurement results. The sensor’s efficacy is demonstrated with different gases, He, N_2_, Ar, and CO_2_, for thermal conductivity and thermal flow measurements. The sensor can accurately measure the thermal conductivity of various gases, including air, enabling correction of flow rate measurements based on the fluid type. The measured voltage from the thermal conductivity sensor for air corresponds to a calculated thermal conductivity of 0.02522 [W/m·K], with an error within 2.9%.

## 1. Introduction

Thermal flow sensors are used to measure the flow rate of both gases and liquids, and can be divided into three basic categories: anemometric, calorimetric, and time-of-flight. Thermal flow sensors are typically composed of a heater and one or more temperature sensors; and follow a similar working principle, i.e., supplying power to the heater to elevate the temperature, and then measuring the change in temperature distribution over the sensing structure as a measure for the flow rate [1,2,3,4,5,6,7].

Thermal flow sensors have a simple working principle and low fabrication cost. However, they are dependent on the type of the flowing medium, more specifically the thermal properties of the gas or liquid. This means that calibration of these sensors is required whenever the medium changes. Several techniques have been investigated to make thermal flow measurements medium-independent. Many of these involve the use of AC measurements for measuring both the flow velocity and fluid properties. When an AC current with frequency ω is used for Joule heating, this will result in temperature variations at frequency 2ω, and the heater voltage will contain a component at frequency 3ω due to the dependence of the heater resistance on temperature. This technique has been used by Chung et al. [8], who presented tunable AC thermal anemometry to find the phase lag at which an anemometer is the most sensitive to the flow speed, and Heyd et al. [9], who proposed a hot-wire anemometer carrying a sinusoidal electric current and used the 3ω component to detect the conductive-convective exchange coefficient and the gas flow rate. Gauthier et al. [10] used the 3ω technique to measure the thermal conductivity of pure gases, nitrogen, helium, and carbon dioxide at 298 K under atmospheric pressure. The obtained error between the measured thermal conductivities and reference values was less than 5%. Kuntner [11] proposed a micromachined chip that contained a heater and two germanium thermistors placed on a thin silicon nitride membrane. An AC signal was applied to the heater and the resulting temperature variations at the thermistors were measured to determine the thermal conductivity and diffusivity of liquids. The same technique was used by Beigelbeck [12,13] to measure the thermal conductivity and diffusivity of various liquids (water, glycerol, olive oil, silicon oil, and insulating oil) and nitrogen. Hepp et al. [14] used AC excitation to simultaneously determine the flow speed and gas concentration of a binary gas mixture. They applied an AC sinusoidal signal to a heater and measured the amplitude and phase shift at a downstream temperature sensor, which were dependent on the flow speed and thermal conductivity of the flowing gas, respectively. They determined the thermal conductivity with an error of less than 10%. Reyes [15] used AC excitation to measure the voltage of a sensor close to the heater. With increasing frequency, the thermal boundary layer decreases and can be brought down close to the wall. Then, due to the non-slip condition at the wall, the thermal exchange between the heater and sensor becomes independent of the velocity, so the heat transfer is only affected by the physical properties of the gas or fluid and not by the flow. Two physical parameters, thermal conductivity *k* and volumetric heat capacity $\rho c_{p}$, were derived from the phase and amplitude of the third harmonic of the measured AC voltage. The flow rate itself is measured with DC excitation which is dependent on *k* and $\rho c_{p}$. By estimating the fluid properties from the AC measurement, a gas-independent flow rate measurement can be achieved. However, this sensor required a relatively complicated data acquisition procedure to extract the thermal parameters. In [16,17], a thermal flow sensor was used consisting of a heater and upstream and downstream temperature sensors to determine the flow speed and thermal properties of binary gas mixtures. In the proposed technique, the thermal conductivity is obtained by operating a heater at constant power. Next, the flow speed is obtained using constant temperature operation and correcting the result with the measured thermal conductivity. Finally, the measured thermal conductivity and flow speed are used to obtain the volumetric heat capacity by observing the heater’s temperature decrease at a specific flow speed. By using the thermal properties of the gas mixtures obtained from the previous steps, real-time determination of gas concentration is possible. A drawback of the proposed sensor is that it can only be used to determine the gas properties in a specific flow region. The found accuracy was better than ±3%.

Besides AC measurements, a few other techniques have been used to measure thermal parameters of the fluid. Inside micromachined channels, the thermal conductivity has been measured using heated elements with a design optimized to have a low dependency on flow or placed in a dead volume [18,19]. In [20,21,22], a multi-parameter flow measurement system was proposed that consisted of an integrated Coriolis and thermal flow sensor, and a pressure sensor. This integrated system allowed for on-chip measurement of flow rates as well as various physical characteristics such as density, viscosity, specific heat, and thermal conductivity of both gases and liquids, by combining the different sensor signals.

In this paper, a novel design is proposed for a sensor probe that can be used to measure the flow rate in a larger tube. It can simultaneously measure the thermal conductivity by simply using a single heated wire with a DC voltage measurement [23].

The paper is divided into sections as follows. In Section 2, the sensor design and its working principle are presented. Section 3 discusses the simulation results of the sensor in COMSOL Multiphysics. In Section 4, the fabrication process of the proposed sensor is explained. Finally, in Section 5, the results of the flow rate sensor and the thermal conductivity sensor with different gases are discussed.

## 2. Design and Operating Principle

### 2.1. Thermal Flow Sensor

Figure 1a shows a schematic drawing of the sensor chip on a PCB, consisting of three pairs of wires realized at the front and back side of a silicon wafer to form calorimetric flow sensors, see Figure 1b, and a single wire structure to form a thermal conductivity sensor, see Figure 1c. The wires are made of SiRN and Cr/Pt layers. A 3D-printed channel is made and the chip glued on the PCB is inserted into it as shown in Figure 1d. The distance between the wires in the flow sensor is defined by the wafer thickness. Geometrical parameters of the sensor are provided in Table 1. The width of the wire is chosen to be as small as possible within the limits given by the fabrication process. A large length-to-width ratio is chosen to eliminate the heat conduction loss through the wires to the silicon substrate. Figure 2 illustrates the working principle of the hot wire as a thermal flow sensor. With increasing flow velocity, the temperature of the heated wires will decrease, resulting in a change in the electrical resistance of the wires, see Figure 2b. For a single hot wire, the temperature distribution upstream and downstream of the wire can be expressed as follows [24,25]:(1)$$T(x)=\left\{\begin{array}{cc} T_{0}e^{\frac{x+w/2}{l_{c1}}} & x\leq -w/2\\ T_{0} & -w/2<x<w/2\\ T_{0}e^{\frac{x-w/2}{l_{c2}}} & x\geq w/2 \end{array}\right.$$
(2)$$T_{0}=\frac{P}{\pi d_{ch}k_{f}\left(A+B\sqrt{v}\right)}$$
(3)$$l_{c1},l_{c2}=\frac{2D}{v\pm \sqrt{v^{2}+\frac{8D^{2}}{d_{ch}^{2}}}}$$

Here, $T_{0}$ is the temperature of the wire when the flow is zero, *P* is the dissipated power in the wire, $k_{f}$ is the fluid thermal conductivity, *A* and *B* are constants, *v* represents the fluid velocity, $D=\frac{k}{\rho c_{p}}$ [W/m${}^{2}\cdot$J] the thermal diffusivity of the fluid, and $d_{ch}$ is the diameter of the channel. The temperature distribution in the *x* direction along the channel length can be calculated from the linear combination of the temperature distribution of each wire, as shown in Figure 3a. At zero flow rate, the temperature difference between the two wires is zero, and by increasing the flow rate the heat transfer from the upstream wire to the downstream wire results in a temperature difference between the two wires. The temperature of the hot wires at the location of $\pm l_{m}/2$ can be derived to obtain the temperature difference between wires.
(4)$$\Delta T=T\left(l_{m}/2\right)-T\left(-l_{m}/2\right)=T_{0}\left(e^{\frac{l_{m}}{l_{c2}}}-e^{\frac{-l_{m}}{l_{c1}}}\right)$$

Figure 3b shows the temperature difference between the upstream and downstream hot wires as a function of the mass flow rate for four different pure gases.

A Wheatstone bridge is employed to measure the voltage difference between the two wires in the calorimetric flow sensor. Since the ambient temperature changes are equal at both wires, the ambient temperature drift would be eliminated in the output voltage of the Wheatstone bridge. Figure 2c shows the circuit schematic of the Wheatstone bridge. Resistors $R_{1}$ and $R_{4}$ are fixed resistors integrated on the silicon chip. Resistors $R_{2}$ and $R_{3}$ are the sensor wires. In no-flow condition, $R_{1}$ and $R_{4}$, and $R_{2}$ and $R_{3}$ have the same value, so the output signal of the Wheatstone bridge $V_{b}$ is zero. When flow is applied, the heat will be transferred from the upstream wire to the downstream one. Therefore, there will be a positive or negative (depending on the flow direction) output voltage signal as a result of the temperature difference between the two wires $R_{2}$ and $R_{3}$. Three pairs of wires are used to measure the flow profile along the z direction of the tube. In a circular tube, the flow velocity is maximum at the center of the tube due to the parabolic flow profile and decreases closer to the tube wall.

### 2.2. Thermal Conductivity Sensor

An additional wire is suspended above a shallow V-groove cavity for thermal conductivity measurement. Figure 4a shows a schematic cross-sectional drawing of the V-groove with the under-etched beam. The wire has the same dimensions as the wires in the thermal flow sensors. The temperature of this wire is dominated by the thermal conductivity of the gas inside the cavity and largely independent of the flow velocity. It is independent of the flow velocity since the wire is close to the channel wall where the velocity is very low, and the flow is perpendicular to the chip surface, which locally blocks the flow. Hence, by monitoring the voltage drop over the wire at a constant heating current, the thermal conductivity of the gas can be detected. Figure 4b shows the circuit schematic for the thermal conductivity sensor. The stationary temperature profile $T\left(x\right)=T_{wire}-T_{a}$ along the length of the beam is defined by the following differential equation [18]: (5)$$\frac{1}{R_{b}^{{}^{\prime }}l^{2}}\frac{\partial^{2}T(x_{n})}{\partial x_{n}^{2}}-G_{f}^{{}^{\prime }}T\left(x_{n}\right)=-P^{{}^{\prime }}$$
(6)$$G_{f}^{{}^{\prime }}=k_{f}\frac{w_{V}}{d_{eff}}$$
(7)$$R_{b}^{{}^{\prime }}=\frac{1}{k_{b}A}$$

In the above equations, $x_{n}$ is the dimensionless normalized position along the wire ranging from −0.5 to +0.5, *l* is the length of the wire, $w_{V}$ and $d_{eff}$ are the width and effective depth of the cavity which is approximately half of the depth of the cavity, $k_{f}$ and $k_{b}$ are the thermal conductivities of the fluid and beam, respectively, and *A* is the cross-sectional area of the beam. $P^{\prime }$ is the electrical line power in [W/m] dissipated at position $x_{n}$, $G_{f}^{{}^{\prime }}$ is the line conductance through the gas in [W/(K·m)], and $R_{b}^{{}^{\prime }}$ is the thermal line resistance of the beam in [K/(W·m)]. The solution for the differential equation is given by [18]: (8)$$\frac{T(x_{n})}{P^{{}^{\prime }}}=\frac{1}{G_{f}^{{}^{\prime }}}\left(1-\frac{\cosh(x_{n}l\sqrt{R_{b}^{{}^{\prime }}\;G_{f}^{{}^{\prime }}})}{\cosh(0.5l\sqrt{R_{b}^{{}^{\prime }}\;G_{f}^{{}^{\prime }}})}\right)$$

Figure 5 shows the calculated temperature distribution along the normalized wire length. The average temperature is calculated by integrating the temperature over the length of the wire: (9)$$T_{ave}=\frac{1}{l}\int_{0}^{l}T\left(x\right)\;dx$$

Then, the voltage over the wire is given by: (10)$$V=R_{a}\left(1+\alpha T_{ave}\right)I$$

$R_{a}$ is the ambient temperature, α is the coefficient temperature of resistance of the beam, and *I* is the dc input current. The calculated voltage for different gases is listed in Table 2.

## 3. Simulation

COMSOL Multiphysics is used to simulate the temperature distribution and temperature difference of the heated wires. Three types of physics, laminar flow, heat transfer in solids and fluids, and electric currents, are used. The mass flow rate is varied from 0 to 50 g/h with no viscous stress and zero pressure and suppressing backflow. The side walls are set to no slip. The Si substrate is set to ambient temperature. As a proof of concept, a simple model is used to simulate the thermal flow sensor and thermal conductivity sensor. For both sensors, a DC current of 5 mA is used to heat up the wire.

Figure 6a shows the thermal conductivity sensor temperature distribution at the mass flow rate of 50 g/h. Due to the gases’ different thermal conductivities, the amount of the heat transferred from the wire to the heat sink should be different. The width of the V-groove cavity underneath the wire is 40 μm. The COMSOL simulation was also used to find the width of the V-groove cavity. A smaller width for the V-groove decreases the sensitivity and a larger width will make the sensor flow dependent. The temperature of the wire as a function of mass flow rate for four different gases (He, N_2_, CO_2_, and Ar) is simulated, see Figure 6b. As can be seen, the temperature of the wire for different gases is distinctive due to their different thermal conductivity.

The same wires as in the thermal conductivity sensor are used for the thermal flow sensor. The distance between the wires is defined by the thickness of the silicon wafer which is 380 μm. Figure 7a,b show the temperature distribution around the upstream and downstream wires with the gas Ar at 50 g/h and the temperature difference between the wires for four gases versus the mass flow rate from 0 to 50 g/h, respectively.

## 4. Fabrication

Figure 8 shows a summary of the fabrication process and photograph of the released chip. To fabricate the sensor first, a layer of 1 μm SiRN is deposited by LPCVD, (Figure 8a). Then, a 20 nm Cr adhesion layer and 200 nm Pt layer are deposited and etched by sputtering and IBE etching, respectively, to pattern the wires and metal traces, (Figure 8b–d). The IBE etching step is performed twice with two different masks. The first step is for transferring the metal pattern, the second one to narrow the beam width and define the pattern in the SiRN layer, (Figure 8e,f). In Figure 8g, SiRN is etched by plasma etching to open the window for etching the Si. All these steps are repeated for the backside of the wafer to have the same structure on both sides. Finally, Si is etched by KOH (KOH 1:3 DI-water) to realize a cavity inside the wafer between the two wires, (Figure 8h). Figure 9a,b show microscope pictures of the released chip.

Because of the large aspect ratio of the beam length to the channel width, a good alignment to the <111> crystal orientation is required to minimize the under etch. Therefore, a Vangbo mask is used to obtain the required crystal orientation [26]. A layer of 150 nm LPCVD deposited SiRN is used as the etch mask to transfer the Vangbo pattern. By wet anisotropic etching in KOH (25 wt.% −75 ${}^{\circ }$C—etch time 10–15 min), the crystallographic orientation of a silicon wafer can be found easily and within an error of ±0.05 degrees. Figure 9c shows the Vangbo mask and the etch structure of the silicon. The second structure shows a perfect alignment while the right structure is not aligned parallel to the <111> crystal orientation, resulting in the non-symmetric under etch.

## 5. Results and Discussion

### 5.1. Thermal Conductivity Sensor

Figure 10 shows the experimental setup, consisting of a Coriolis mass flow controller (MFC), pressure controller (PC), digital voltmeter, and a 3D-printed tube with the chip inside it.

In the thermal conductivity sensor, a single wire suspended above the V-groove cavity was used to measure the thermal conductivity. The wire is fed by a DC current of 5 mA. The gas is applied by a mass flow controller to the 3D-printed channel, and the voltage of the wire is measured by a digital multimeter while the gas passes over the wire. Figure 11a compares the measured and simulated voltage drop over the thermal conductivity sensor as a function of mass flow rate for He, N_2_, Ar, and CO_2_. These gases were chosen specifically because of their different thermal conductivities. As can be seen, the measured voltage depends strongly on the thermal conductivity of the gas. At low flow levels, we see a small influence of the flow, because the outside air enters the tube mixing with the pure gas. Figure 11b compares the thermal conductivity as a function of the measured voltage, the theoretically derived voltage using Equation (Equation 10), the voltage obtained by simulation, and the curve fitted to the measurement result. The measured voltages correspond well with the theoretical response. The fitted equation to the measurement result is: (11)$$\frac{1}{k}=aV_{beam}+b$$

Here, *a* and *b* are the fitting coefficients that are expressed in Table 3. From the Equation (Equation 11), the unknown gas thermal conductivity can be found by measuring the voltage of the wire; then, the output signal of the flow sensor can be adjusted with the measured thermal conductivity to be able to measure the flow rate independent of the gas type [23].

### 5.2. Thermal Flow Sensor

There are three pairs of wires on the sensor chip which form three calorimetric flow sensors. Each of the three wire pairs forms half of a Wheatstone bridge. The other half of the bridge is formed by fixed on-chip resistors (see Figure 2c). The sensor wires are heated by the DC voltage that is applied to the bridge, which is 2 V. At room temperature, all wires have a resistance of approximately 300 Ω. A difference in temperature between the sensor wires will result in an output voltage.

Figure 12a shows the measured output voltages of the three sensors as a function of the volumetric flow rate of N_2_. However, the sensitivity to the location of the sensor wires is low. Four different gases, He, N_2_, Ar, and CO_2_ are also used for thermal flow measurement, see Figure 12b. This shows that the voltage of the Wheatstone bridge is dependent on the type of the gas due to the gases’ different thermal properties $\left(V_{bridge}=f\left(k,\rho c_{p},Q\right)\right)$. The sensitivity of the upper calorimetric flow sensor for each gas is shown in Table 4. As is expected due to the highest $\rho c_{p}$ of CO_2_, it has the highest sensitivity, and He has the lowest sensitivity because of its lowest $\rho c_{p}$, see Table 5. At higher flow rates, due to the turbulent flow (Re>2000), the output voltage becomes noisy. The calculated bridge voltage as a function of the thermal properties of the gas and flow rate is obtained as:(12)$$V_{bridge}=G\frac{\rho c_{p}}{k^{1.7}}Q$$

*G* is a constant value of 5.46 ×$10^{-4}$$\left[\frac{W^{1.7}}{K^{0.7}\cdot m^{1.7}\cdot A}\right]$. Equation (Equation 12) is empirically derived from Figure 12b. Figure 13 illustrates the applied mass flow rate versus the measured flow rate calculated by Equation (Equation 12) for five various pure gases. At low flow rates due to the measurement limitations, gases with a low density such as He are mixed with air in the tube, resulting in a deviation between the measured and applied flow rates.

### 5.3. Simultaneous Measurement of Flow Rate and Thermal Conductivity

The thermal conductivity of the unknown gas can be measured by the proposed thermal conductivity sensor by measuring the voltage of the wire and finding the thermal conductivity from Equation (Equation 11). In this experiment, a test gas air is applied to the sensor through a Mini-Coriolis MFC. The experiment is conducted in five cycles, with the flow rate ranging from 0 to 50 g/h in 50 steps. A digital multimeter measured a 1.728 V DC voltage, which is used in Equation (Equation 11) to determine the thermal conductivity of the gas (0.02522 [W/m·K]), which is within 2.9% of the theoretical value listed in Table 5. This deviation is similar to that obtained in [17]. Then, by applying the thermal conductivity and the corresponding volumetric heat capacity from the Table 5 into Equation (Equation 12), the flow rate can be determined. Figure 14 shows the bridge voltage as a function of the measured and applied flow rate for pure air. A back-and-forth flow rate ranging from 0 to 0.7 L/min in five cycles is applied to the sensor and the voltage of the bridge is measured, and the bridge voltage is also used in Equation (Equation 12) to calculate the flow rate. It proves that the test gas, pure air, is successfully detected by the thermal conductivity sensor, and simultaneously used to measure the flow rate using the empirical Equation (Equation 12).

## 6. Conclusions

In this paper, a novel potentially gas-independent thermal flow sensor chip is presented. It consists of three pairs of wires used as calorimetric flow sensors to measure the flow profile and flow direction inside a flow channel, and a flow-independent thermal conductivity sensor that detects the type of gas through a simple DC voltage measurement. Different gases are used for the thermal conductivity measurement and the measured output voltage corresponds well with the theoretical model. Pure air is utilized as a test gas, and its thermal conductivity is obtained from the measured voltage using the theoretical model, Equation (Equation 11). An empirical relation was found for the thermal flow sensor which relates the thermal properties of the gas and the flow rate. This relation was used to measure the flow rate by compensating for the thermal conductivity.

## Figures and Tables

**Figure 1 micromachines-14-01280-f001:**
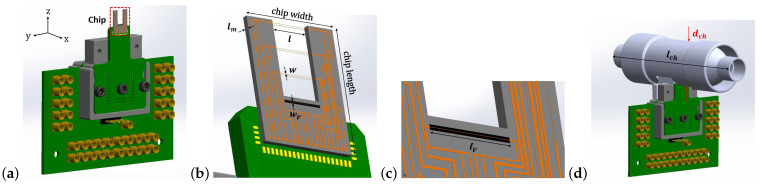
(**a**) Schematic drawing of the sensor on PCB, (**b**) chip consisting of three pairs of wires realized at the front and back side of a silicon wafer to form calorimetric flow sensors, and thermal conductivity sensor, (**c**) a close-up image of thermal conductivity sensor with the cavity underneath, (**d**) the chip glued on the PCB inserted into the tube.

**Figure 2 micromachines-14-01280-f002:**
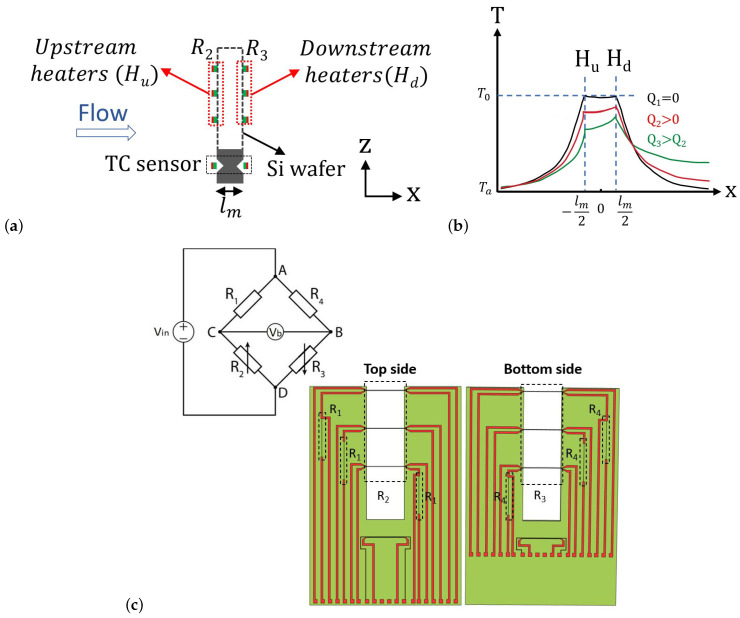
Illustration of the working principle of the thermal flow sensor. (**a**) Cross-section view of the sensor chip. (**b**) The temperature profile in *x* direction. By increasing the flow rate, the temperature difference between the upstream and downstream wires increases. (**c**) Circuit schematic of the Wheatstone bridge (The bridge is fed by 2V) and the top and bottom sides of the wafer. $R_{1}$ and $R_{4}$ are the fixed resistors with constant temperature and $R_{2}$ and $R_{3}$ are the variable transistors.

**Figure 3 micromachines-14-01280-f003:**
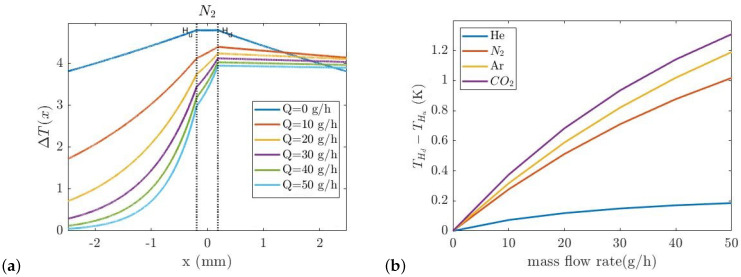
(**a**) Temperature distribution along the length of the channel in *x* direction for N_2_, (**b**) The temperature difference between upstream and downstream hot wires ($T_{H_{d}}-T_{H_{u}}$) as a function of mass flow rate in thermal flow sensor for He, N_2_, Ar, and CO_2_.

**Figure 4 micromachines-14-01280-f004:**
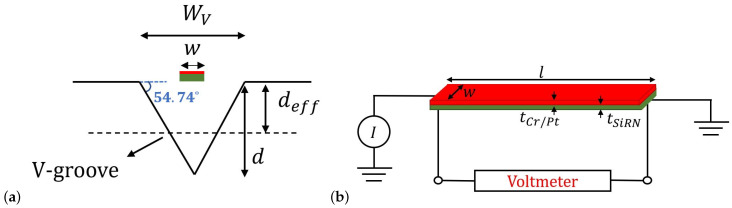
(**a**) Schematic cross-sectional drawing of V-groove with under etched beam. (**b**) Circuit schematic of the thermal conductivity sensor. A 5 mA DC current is applied to the wire, and the voltage of the wire is measured with a digital multimeter.

**Figure 5 micromachines-14-01280-f005:**
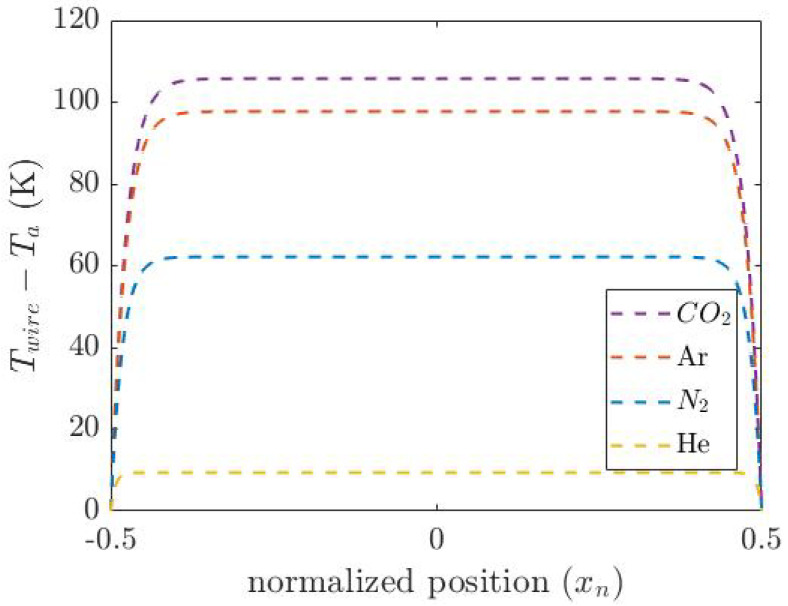
Calculated temperature distribution along the beam of the thermal conductivity sensor for four different gases (He, N_2_, Ar and CO_2_).

**Figure 6 micromachines-14-01280-f006:**
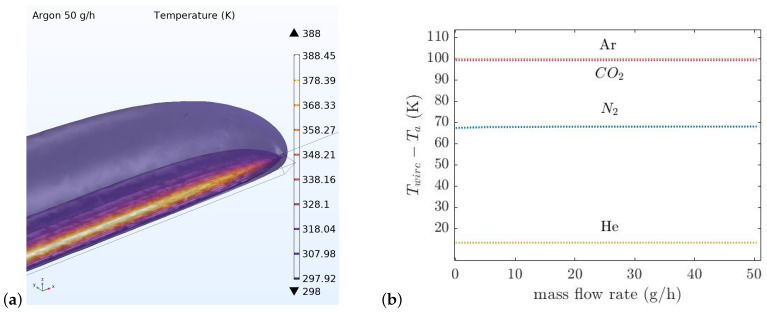
Simple model of thermal conductivity sensor simulated in COMSOL Multiphysics. (**a**) The temperature distribution at 50 g/h. (**b**) The average temperature of the thermal conductivity sensor’s wire as a function of mass flow rate.

**Figure 7 micromachines-14-01280-f007:**
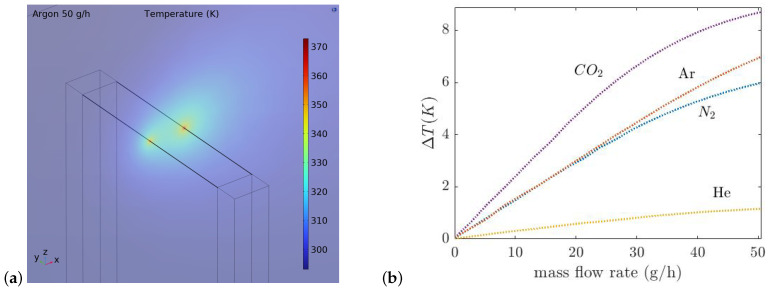
Simple model of thermal flow sensor simulated in COMSOL Multiphysics. (**a**) The temperature distribution at 50 g/h. (**b**) The temperature difference between the wires for four gases versus the mass flow rate from 0 to 50 g/h.

**Figure 8 micromachines-14-01280-f008:**
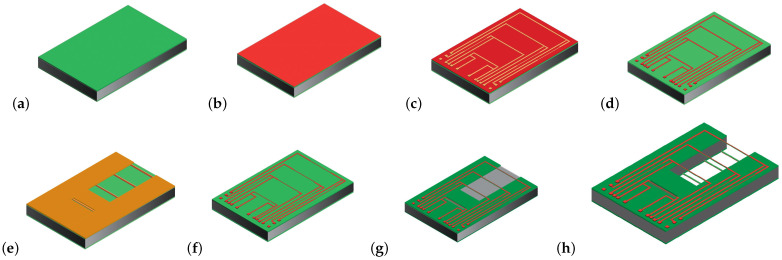
Overview of the fabrication steps. (**a**) LPCVD of 1 μm SiRN, (**b**) sputter a layer of Cr/Pt (20 nm/200 nm), (**c**,**d**) IBE etching of Cr/Pt, (**e**–**g**) SiRN layer is etched by plasma etching to open the window for etching the Si. Then, all these steps are repeated for the backside of the wafer. (**h**) Finally, Si is etched by KOH (KOH 1:3 DI-water) to realize a V-groove and a cavity inside the wafer between the wires.

**Figure 9 micromachines-14-01280-f009:**
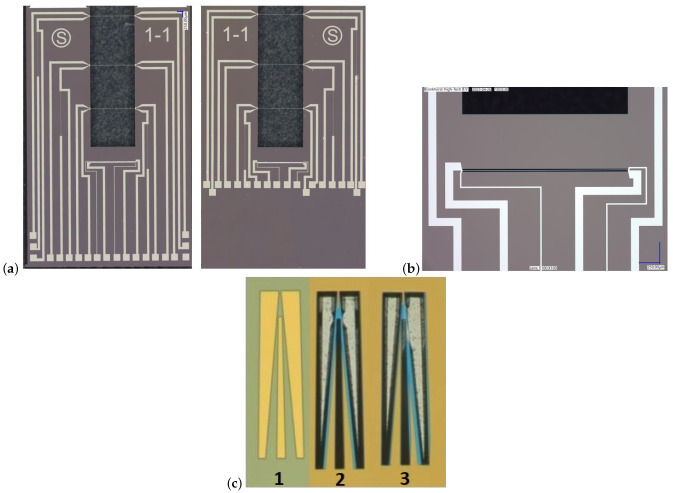
Microscope photograph of the released sensor. (**a**) shows the sensor chip on each side of the silicon wafer. (**b**) The zoom-in picture from the thermal conductivity sensor. (**c**) Vangbo mask and the etch structure of the silicon; (**1**) Vangbo mask, (**2**) Symmetric under-etch, and (**3**) Misaligned etch structure.

**Figure 10 micromachines-14-01280-f010:**
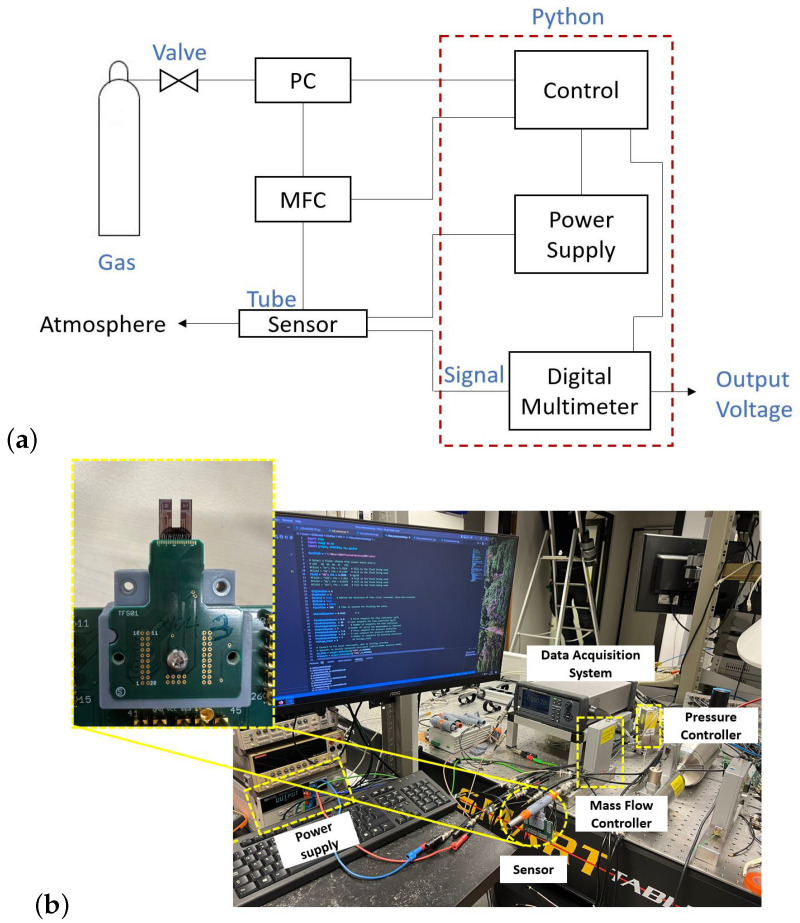
(**a**) Schematic drawing of the measurement setup. (**b**) Experimental setup. The setup consists of a pressure and flow controller for providing the gas flow to the 3D-printed flow tube, an excitation system for heating the wire, and a data acquisition system for reading the voltage of the wire. The acquisition is implemented in a Python program.

**Figure 11 micromachines-14-01280-f011:**
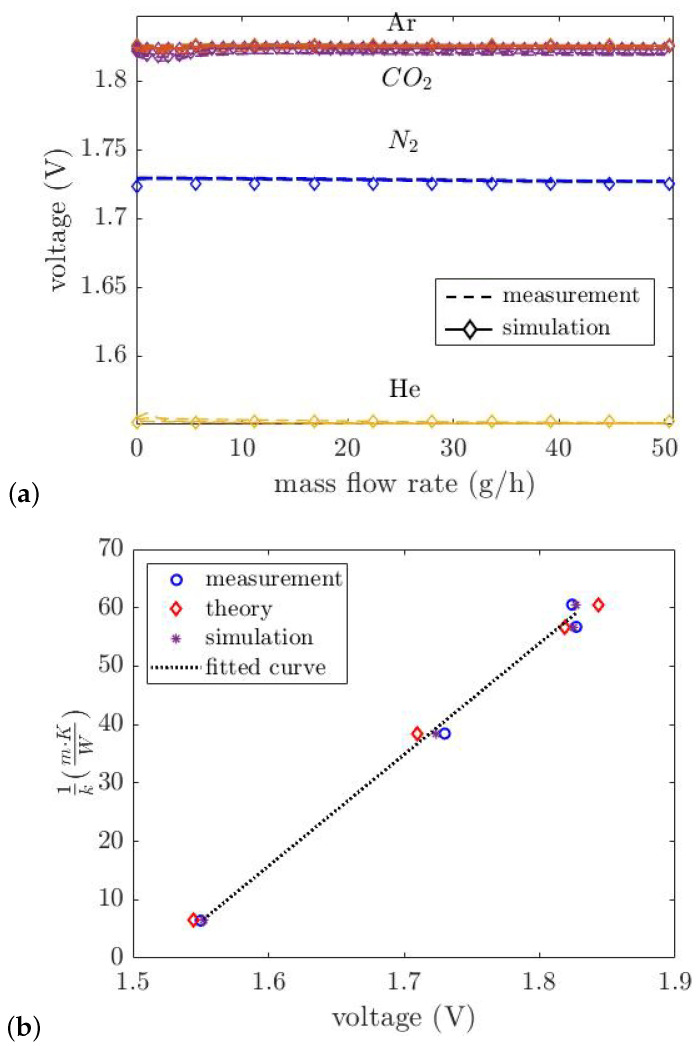
(**a**) Measured and simulated voltage of the thermal conductivity sensor versus mass flow rate for pure gases He, N_2_, Ar and CO_2_. The wire is heated with a constant current value of 5 mA, and the voltage of the wire is measured. (**b**) Theory simulation and measurement results comparison.

**Figure 12 micromachines-14-01280-f012:**
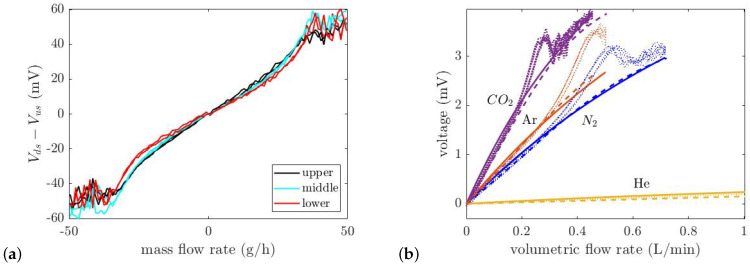
(**a**) Difference voltage of upstream and downstream wires as a function of volumetric flow rate (L/min) for three pairs of wires with N_2_. (Each wire is fed by a DC current of 5 mA). (**b**) Comparison between the measured, theoretically calculated, and simulation voltage of upper sensor versus volumetric flow rate for four pure gases (He, N_2_, Ar and CO_2_). At lower flow rates, the theory and simulation results correspond to the measurement.

**Figure 13 micromachines-14-01280-f013:**
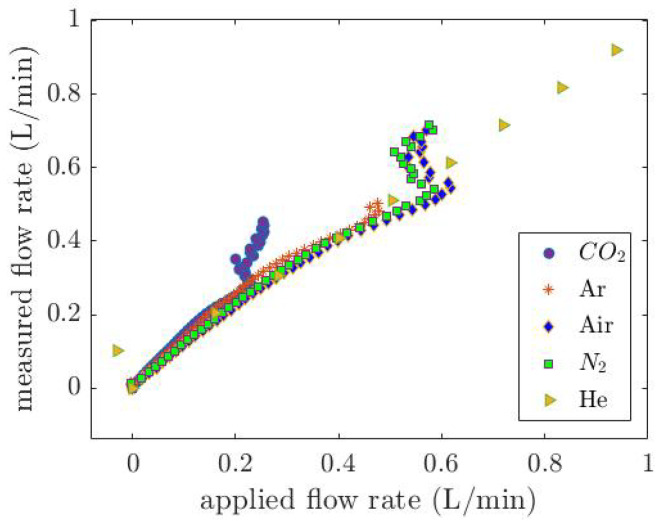
The applied mass flow rate versus the measured flow rate calculated by Equation (Equation 12). The deviation between the measured and applied flow rates at lower flow rates is due to the measurement limitations that result in the air entering into the tube and mixing with the He.

**Figure 14 micromachines-14-01280-f014:**
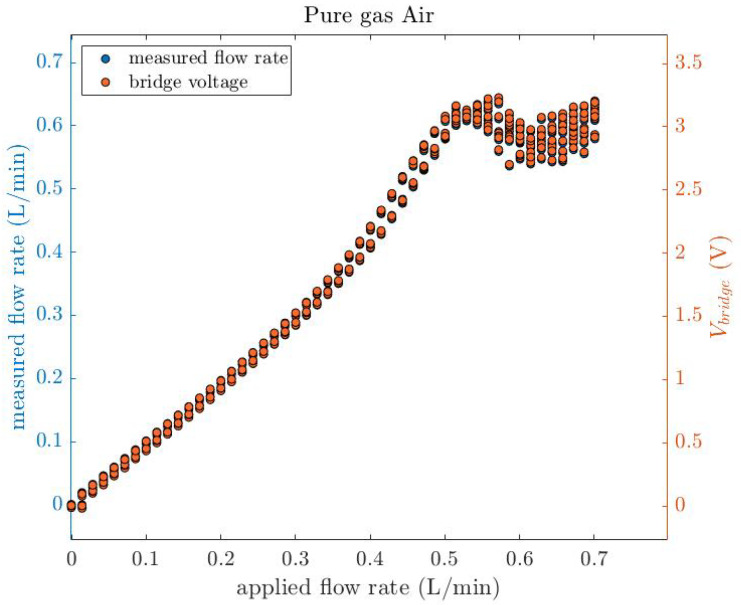
The bridge voltage as a function of the measured and applied flow rate for pure air. A back-and-forth flow rate ranging from 0 to 0.7 L/min in five cycles is applied to the sensor and the voltage of the bridge is measured, and the bridge voltage is also used in Equation (Equation 12) to calculate the flow rate.

**Table 1 micromachines-14-01280-t001:** Geometrical parameters used for the sensor.

Parameter	Term	Value
Beam length	*l*	2000 μm
Beam width	*w*	5 μm
Silicon wafer thickness	$l_{m}$	380 μm
V-groove width	$W_{V}$	40 μm
V-groove length	$l_{V}$	2000 μm
Chip width	-	7500 μm
Chip length	-	11,500 μm
Channel diameter	$d_{ch}$	2 cm
Channel length	$l_{ch}$	10 cm
V-groove depth	*d*	28 μm
Effective V-groove depth	$d_{eff}$	15 μm
Input current	*I*	5 mA
Ambient temperature	$T_{a}$	293 K
Ambient resistance	$R_{a}$	300 Ω

**Table 2 micromachines-14-01280-t002:** The theoretically calculated voltage of the beam using Equation (Equation 10).

Gas	He	N_2_	Ar	CO_2_
Voltage (V)	1.55	1.71	1.81	1.83

**Table 3 micromachines-14-01280-t003:** The fit parameters for the thermal conductivity wire.

Coefficients	a [m·K/A]	b [m·K/W]
	189.4278	−287.6934

**Table 4 micromachines-14-01280-t004:** The calculated sensitivity of the thermal flow sensor for various gases.

Gas	CO_2_	N_2_	Air	Ar	He
Sensitivity (mV/L/min)	10.41	5.91	5.09	4.96	0.31

**Table 5 micromachines-14-01280-t005:** The physical properties of the five various pure gases [27].

Gas Physical Properties	CO_2_	Ar	Air	N_2_	He
Thermal conductivity (W/m·K)	0.01652	0.01763	0.02598	0.026	0.1554
Density (kg/m${}^{3}$)	1.84	1.662	1.189	1.165	0.1664
Heat capacity (J/kg·K)	846.8	521.9	1006	1043	5196

## Data Availability

Not applicable.

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
