# Peer review of "Thermal Flow Meter with Integrated Thermal Conductivity Sensor"

_micromachines, 2023, doi:10.3390/mi14071280_

Round 1

Reviewer 1 Report

The manuscript entitled “Thermal Flow Meter With Integrated Thermal Conductivity Sensor” described a gas-independent thermal flow sensor chip that contains three calorimetric flow sensors. The topic is interesting, and the experimental data supported the conclusions. But the authors should address the following issues before publish on this journal. The detail comments are as follows:

1.     The abstract part should be rewritten. The authors should give one or two sentences to introduce the meaning of this research at the beginning of this paragraph. Then the authors should summarize the advantages of the sensor chip compared with other works. More accurate experimental data should be given rather than simple state your chip could effectively work.

2.     In the introduction part, I suggest the authors to well summarize the published work. The accuracy and/or the disadvantages of others’ works. The references are not enough in the introduction part. Please cite more reference, especially the newly references.

3.     How do you determine the dimensions of the chip? Have you done some optimization?

4.     Why do you choose Ar and He as the gas flow rather than O2 and H2O, which are more common?

Author Response

Reviewer 1

The manuscript entitled “Thermal Flow Meter With Integrated Thermal Conductivity Sensor” described a gas-independent thermal flow sensor chip that contains three calorimetric flow sensors. The topic is interesting, and the experimental data supported the conclusions. But the authors should address the following issues before publish on this journal. The detail comments are as follows:

1)The abstract part should be rewritten. The authors should give one or two sentences to introduce the meaning of this research at the beginning of this paragraph. Then the authors should summarize the advantages of the sensor chip compared with other works. More accurate experimental data should be given rather than simple state your chip could effectively work.

Thank you very much for your comment. We have now rewritten the abstract to reflect more clearly the contents of the paper.

2) In the introduction part, I suggest the authors to well summarize the published work. The accuracy and/or the disadvantages of others’ works. The references are not enough in the introduction part. Please cite more reference, especially the newly references.

We have rewritten and extended the introduction and added more references to earlier published work.

3) How do you determine the dimensions of the chip? Have you done some optimization?

The dimensions of the wires are limited by the fabrication process. What is important is to choose a large length to width ratio of the wires to eliminate the heat conduction loss through the wires to the silicon substrate. The COMSOL simulations were also used to find the dimension of the v-groove cavity. A smaller width for the v-groove decreases the sensitivity and a larger width will make the sensor flow dependent. This information has now been included in the manuscript, in lines 91-93 and lines 163-166.

4) Why do you choose Ar and He as the gas flow rather than O2 and H2O, which are more common?

These gases were chosen specifically because of their different thermal conductivities. Pure oxygen is not allowed in our lab. We added a motivation for the choice of the gases in line 205-206.

Reviewer 2 Report

1)  There is no information given in the introduction concerning the accuracy of the competing flow measurement techniques covered in references 8-18, which is needed to establish state of the art for such measurements.

Accuracy of the proposed sensor is also missing in the later discussion, for example that the measured thermal conductivity for air given on line 205 is 0.02487 compared to the FLUIDAT value in Table 5 of 0.02598, which gives an error of 4%. This can be compared to relative error of  <2% reported for a number of gases in C J Hepp et al., Flow rate independent sensing of thermal conductivity in a gasstream by a thermal MEMS-sensor – Simulation and experiments, Sensors and Actuators A 253 (2017) 136-145.

2)  Related to the above is that line 200 brings up the unknown gas, which is what sensor science is concerned with, but the following experiment describes that the gas used is known to be air. The performance of the sensor needs to be validated in a blind test where one person controls the flows of gases in random order and another determines identities and flow rates with reporting of the resulting accuracies. Inclusion of gases not listed in table 5 would raise the degree of difficulty.

3)  It's not readily obvious how the voltages shown  in Fig 12a are related to those in 5b of reference 19.  The reference shows (according to caption a) bridge voltages for three pairs of wires on a scale of -4 to +4 mV. Fig 12a shows the voltage difference between the upstream and downstream wires for the three pairs on a scale of -60 to +60 mV.

4)  It may just be that in simulations everything is relative, but the average temperature of the sensor wire in He shown in Fig 6b is 10 K, not that far from absolute zero, and the temperature distribution in Fig 7a is a more realistic 300 to 370.   

Author Response

Reviewer 2

There is no information given in the introduction concerning the accuracy of the competing flow measurement techniques covered in references 8-18, which is needed to establish state of the art for such measurements.

Thank you very much for your comments. We have rewritten and extended the introduction with more references to previous work and a more quantitative comparison of the obtained accuracy.

Accuracy of the proposed sensor is also missing in the later discussion, for example that the measured thermal conductivity for air given on line 205 is 0.02487 compared to the FLUIDAT value in Table 5 of 0.02598, which gives an error of 4%. This can be compared to relative error of  <2% reported for a number of gases in C J Hepp et al., Flow rate independent sensing of thermal conductivity in a gasstream by a thermal MEMS-sensor – Simulation and experiments, Sensors and Actuators A 253 (2017) 136-145.

Unfortunately, there was a small error in the fit parameters. After repeating the fit we found a thermal conductivity of air of 0.02522, so that the error compared to the value from FLUIDAT is 2.9%. This error is now also added in the manuscript, in lines 244-247.

2)  Related to the above is that line 200 brings up the unknown gas, which is what sensor science is concerned with, but the following experiment describes that the gas used is known to be air. The performance of the sensor needs to be validated in a blind test where one person controls the flows of gases in random order and another determines identities and flow rates with reporting of the resulting accuracies. Inclusion of gases not listed in table 5 would raise the degree of difficulty.

Indeed, we agree that a blind test would have been ideal. Thank you very much for this suggestion. Unfortunately, at this moment our lab is not equipped to do such an experiment without the person doing the measurement being able to see which gas is used. This is the reason why we chose a gas that was not used in the calibration for this test.

3)  It's not readily obvious how the voltages shown  in Fig 12a are related to those in 5b of reference 19.  The reference shows (according to caption a) bridge voltages for three pairs of wires on a scale of -4 to +4 mV. Fig 12a shows the voltage difference between the upstream and downstream wires for the three pairs on a scale of -60 to +60 mV.

In reference 23 a different Wheatstone bridge was used with much smaller values for the fixed resistances. Unfortunately, this is not mentioned clearly in this reference. In the manuscript the fixed resistors in the Wheatstone bridge are identical to the sensor wires, see lines 221-222, resulting in higher resistance values and a larger output voltage.

4)  It may just be that in simulations everything is relative, but the average temperature of the sensor wire in He shown in Fig 6b is 10 K, not that far from absolute zero, and the temperature distribution in Fig 7a is a more realistic 300 to 370.

Thank you for noticing this. The figures are correct, but the y-axis label in Fig. 6b was indeed confusing. We changed the label such that it is clear that it is the temperature of the wire minus the ambient temperature.

Round 2

Reviewer 1 Report

The authors have addressed my questions. I recommend it to publish in this journal.

Reviewer 2 Report

2)  The blind test would be one person doing the measurements by controlling the flows and identities of the gases used, the other person (not present during the measurements) would determine the flows and identities from the data.

As it stands now there is no evidence presented that the sensor can determine gas flows and identities. This should disqualify publication in a sensor journal, but not necessarily in Micromachines if the editor is satisfied that the scope and aims have been satisfied.